# Machine Translation of Electrical Terminology Constraints

**Zepeng Wang** [1,2]**, Yuan Chen** [3] **and Juwei Zhang** [1,2,]*

1   School of Electrical Engineering, Henan University of Science and Technology, Luoyang 471023, China; 210321061704@stu.haust.edu.cn
2   Henan Province New Energy Vehicle Power Electronics and Power Transmission Engineering Research Center, Luoyang 471023, China
3   School of Foreign Languages, Henan University of Science and Technology, Luoyang 471023, China; 9903671@haust.edu.cn
*   Correspondence: juweizhang@haust.edu.cn

**Abstract:** In practical applications, the accuracy of domain terminology translation is an important criterion for the performance evaluation of domain machine translation models. Aiming at the problem of phrase mismatch and improper translation caused by word-by-word translation of English terminology phrases, this paper constructs a dictionary of terminology phrases in the field of electrical engineering and proposes three schemes to integrate the dictionary knowledge into the translation model. Scheme 1 replaces the terminology phrases of the source language. Scheme 2 uses the residual connection at the encoder end after the terminology phrase is replaced. Scheme 3 uses a segmentation method of combining character segmentation and terminology segmentation for the target language and uses an additional loss module in the training process. The results show that all three schemes are superior to the baseline model in two aspects: BLEU value and correct translation rate of terminology words. In the test set, the highest accuracy of terminology words was 48.3% higher than that of the baseline model. The BLEU value is up to 3.6 higher than the baseline model. The phenomenon is also analyzed and discussed in this paper.

**Keywords:** character level word segmentation; encoder residual connection; machine translation in electrical domain; terminology phrase dictionary

## 1. Introduction

Machine translation [1] is the process of converting source code language to target language by using a computer. As a cross-language communication tool, machine translation plays an increasingly important role in people's daily lives and has become one of the hot contents in the research field of natural language processing [2,3]. In recent years, deep learning [4] technology has made major breakthroughs, and neural machine translation [5] that integrates deep learning technology has gradually replaced statistical machine translation [6] with excellent translation quality, becoming the mainstream machine translation method in academia and widely used in industry.

In the past 70 years of development history, the methodology of machine translation has undergone several major changes. From rule-based machine translation [7], to statistical machine translation [6] during the beginning of the big data era, to today's neural machine translation, the academic research on machine translation technology has never been interrupted. Nowadays, machine translation in the general-purpose domain with more abundant parallel corpus resources has achieved good research results, which has brought great convenience to translation work such as language communication in daily human life. However, the neural machine translation system has the problem of insufficient translation of low-frequency words. Terminology words in specialized domain literature are a kind of low-frequency word carrying important professional information. Whether it is translated correctly or not often has an important influence on the credibility of the translation text [8].

In the current social and economic development, electricity involves all aspects of our lives. Machine translation for the electrical field is of great significance in promoting the progress and development of the electrical industry. There are many terminologies in the literature of the electrical field, and the current translation model often fails to give the correct translation. Therefore, the research on machine translation in the electrical field is of great significance.

The main contributions of this paper are as follows.

This paper takes the parallel corpus in the electrical domain as the data set to construct the terminology phrase dictionary; based on this, three schemes are given to use the term phrase dictionary to constrain the translation model to improve the translation effect of the model. Scheme 1 replaces the terminology phrases of the source language; Scheme 2 uses the residual connection at the encoder end after the terminology phrase is replaced; Scheme 3 uses a segmentation method of combining character segmentation and terminology segmentation for the target language and uses an additional loss module in the training process. The three schemes are superior to the baseline model in terms of BLEU value and correct translation rate of term words. The results show that the proposed method in this paper can make the translation model effectively learn the knowledge of the external terminology phrase dictionary and improve the translation quality of machine translation in the field of electrical engineering.

## 2. Related Work

Whether the terminology in the professional field is translated accurately is very important in practical application. Using external knowledge to constrain translations is one of the ways to improve the quality of terminology word translations. In order to take advantage of pre-defined translations, Crego et al. [9] used placeholder tags to replace named entities on both the source and target language end, allowing the model to learn translation placeholder tags to translate these named entity words. For example, the i-th named entity in the source language sentence is replaced with 'Tag$_{(i)}$', and at the output, the placeholder markers in the output will be replaced with a pre-specified translation. The disadvantage of this method is that the named entities themselves are replaced by labels in the sentences of the input model so that the input sentences lack the semantic information of the named entities, which will affect the adequacy and fluency of the output. In the field of new energy, Dung et al. [10] used two methods of terminology word substitution and splicing to improve the quality of terminology translation in Chinese–English machine translation in the field of new energy, and the results show that the use of labels when replacing splicing will achieve better results. Liu et al. [11] used the combination of placeholder and splicing to improve the accuracy rate of terminology translation in the Chinese–English conference scenes machine translation in three domains: sports, business, and medicine. Dung and Liu [10,11] improved the baseline model with the source language of Chinese and did not solve the problem of word-by-word translation of English phrases and phrase mismatch when the source language was English. Moreover, they did not improve the model structure when using the replacement splicing method, which can cause the loss of label information. In this paper, the method uses a bilingual phrase dictionary to replace the terminology words, which solves the translation problem of English term phrases. Compared with the use of placeholder methods, the method in this paper retains semantic information in data processing and adds residual connection structure at the encoder end, which further improves the use of lexical information of terminology words in the source language.

In the word on English–Chinese translation, Deyi Xiong et al. [12] proposed a method of neural machine translation intervention and fusion using bilingual dictionaries. In order to fuse the translation from bilingual dictionaries into the neural machine translation model, Deyi Xiong et al. [13] proposed three methods: tagging, mixing phrases, and extra embedding. The first two methods implement interventions in the data preprocessing stage and expand the data with words or phrases that appear in the bilingual dictionary in the training data. The additional embedding method is to intervene at the embedding

layer. In addition to word embedding and position embedding, the label signal is input as an additional embedding to the embedding layer. When the three methods are combined, the best results are achieved.

Another mainstream method is to use constraint decoding algorithms in decoding, using external knowledge as hard constraints in decoding. Hokamp [13] proposed a grid beam search algorithm, which uses the pre-specified vocabulary translation as constraints during the beam search process. The cost is to increase the complexity of the standard decoding algorithms and increase the decoding time of the model. Hasler [14] used a multi-stack decoding method and proposed a constrained beam search algorithm, which uses the alignment method to obtain the corresponding source language vocabulary of the target side constraints and constructs a finite state automaton to guide the constraint decoding before decoding. Post and Vilar [15] improved Hokamp's method by proposing a dynamic beam allocation algorithm. This algorithm maintains a single beam of size k during decoding, which limits the decoding complexity and solves the problem that it is difficult to integrate with batch decoding and other operations when using grid beam search algorithm and constrained beam search algorithm for constrained decoding. All three algorithms of constraint decoding make structural changes to the decoder, the complexity is high, and it is difficult to integrate with other methods, which have certain application limitations. In contrast, the method in this paper is simple and efficient, and the main work is completed on the data side to facilitate the integration of more advanced models and structures. The decoding speed of this method is better than that of the constraint decoding method. Because the semantic information contained in Chinese sentences is more complex, the use of hard decoding will destroy the overall semantic integrity of the sentence and affect the fluency of the sentence. The method in this paper cannot guarantee that the translation results contain all the marked words, but it will have a higher overall quality than the constrained decoding method.

Another method of data expansion using external knowledge is knowledge graph fusion. Since knowledge graphs usually contain rich knowledge of named entities and entity relationships, they can be used to improve the named entity translation ability of neural machine translation models. The knowledge graph contains a large number of named entities that do not appear in the machine translation model. These entities can be called extra-domain entities relative to the machine translation model. On the contrary, entities that appear in both the training data and the knowledge graph can be called intra-domain entities. Therefore, the translation of intra-domain entities can be used to guide the translation of extra-domain entities. Zhao et al. [16] used the knowledge representation learning method to learn the source language knowledge graph and the target language knowledge graph to obtain the vector representation of the source language entity and the target language entity. The entity translation pair in the parallel sentence pair is extracted as the seed entity translation pair, and it is used as the anchor point. The vector representation of the source language entity and the target language entity is mapped to the same semantic space. After calculating the semantic distance between them, the translation of the extra-domain entities is predicted to constitute the derived entity pairs. When the semantic distance between the derived entity pair and the seed entity pair is less than the preset threshold of λ, the derived entity pair is replaced by the seed entity pair in the parallel sentence pair to generate a pseudo-bilingual sentence pair. Finally, the pseudo-bilingual data is merged with the original data to complete the fusion of the knowledge graph.

Data augmentation is a general term for a class of methods that increase the diversity of training data by changing existing data or synthesizing new data [17]. Sennrich R et al. [18] proposed a reverse translation method, which uses the monolingual data from the target language to reversely translate into the source language and combines it into pseudo-parallel data and the original parallel data to train the forward translation system. Currey et al. [19] enhanced the training data by copying the target language sentence to the source language, resulting in an enhanced training corpus where the source language and the target language contain the same sentence. The enhanced data proved to improve

translation performance, especially for the same proper nouns and other low-frequency words in the source and target languages.

In recent years, there have been many new studies on the fusion of prior knowledge and the enhancement of key information in text. Wu et al. [20] used BERT to extract context features and proposed three ways to fuse context sequences with source language sequences. Hu et al. [21] proposed that accurate and complete translation of key information in the text can ensure the quality of translation results. Their work is embodied in the fusion of key information in the source language with the source language sentences through the preset encoder, which improves the translation effect of keywords. Ren [22] integrated grammatical information into the Mongolian–Chinese machine translation work. Specifically, the grammatical prior knowledge was integrated into the model through two different grammatical auxiliary learning units preset in the encoder and decoder, which improved the translation quality of the model. Li [23] proposed a method to enhance the factual relationship, which integrates the factual relationship information into the model as a priori knowledge. Specifically, the fact relation mask matrix is constructed in the encoding process as the fact relation representation of the source language. In the decoding process, the actual relation representation and the original representation of the original sentence are merged into the decoder. Chen et al. [24] classified the source language words into content words and function words according to word frequency and designed content word perception NMT. Specifically, the content word is encoded as a new source representation, based on which additional content word context vectors are learned, and the content word-specific loss of the target sentence is introduced. Nguyen et al. [25] used prior alignment to guide the training of the Transformer and tested the influence of different weights. Finally, an 8-head Transformer-HA model using prior alignment with heavier weights to guide multi-head translation was proposed. The disadvantage of this model is that, compared with our method, the prior knowledge will be diluted when faced with long sentences. Peng et al. [26] established the syntactic dependency matrix of each word based on the syntactic dependency tree and integrated quantitative syntactic knowledge into the translation model to guide translation, effectively learning syntactic details and eliminating the dispersion of attention scores.

Machine translation research in low-resource and professional fields has also been a hot topic in recent years. Dungyer et al. [27] created a data set that includes the works of a Croatian-related contemporary poet and the translation of German poetry by two professional literary translators. The research results show the effectiveness of poetry machine translation in terms of special automatic quality indicators. At the same time, Seljan et al. [28] performed manual adequacy and fluency analysis on the machine translation results of poetry under the same data set, which proved the effectiveness of applying machine translation to Croatian–German pairs in the field of poetry. Gašpar et al. [29] proposed a method to test the quality of language texts for terms. When the consistency of terms in the corpus is high, a high HHI score will be obtained. Huang et al. [30] proposed a domain-aware NMT with mask substructure for multi-domain machine translation, which is characterized by the improved model that can be automatically adapted in multi-domain neural machine translation. Specifically, mask substructures are used in both encoders and decoders to capture specific representations in each specific domain, which effectively maintains specific knowledge in each domain and solves the problem of catastrophic forgetting. Yu [31] proposed a method to improve neural machine translation by using similar dictionaries in view of the similarity between Thai and Lao languages. A bilingual similarity dictionary composed of pairs of similar times is established, and an additional similarity dictionary encoder is introduced. The above two methods enhance the specific word representation in low-resource machine translation and improve the effect of machine translation.

In summary, in the existing research on machine translation in the professional field, the main ideas are divided into three categories: (a) Mark the term position in the corpus to allow the model to enhance the learning of the marker position [9–12,20]; (b) Change the

structure of the Transformer encoder to enhance the learning of term information [21–23,25,31]; (c) Use constraint decoding when decoding [13–15]. Although these works have made great contributions to machine translation in the field of low resources, there are still areas for improvement. (a) The class method will affect the semantic information of the original sentence, which in turn affects the overall effect of translation. In addition, marking information only in the corpus will cause information dilution in the process of multiple training of the model. The scheme proposed in this paper uses the target language term phrases when marking and replacing in the corpus, which will not interfere with the original information in content. At the same time, the residual link structure is added to ensure the integrity of semantic information and labels; (b) The class method is generally more complex, and the added information is easily diluted in multiple trainings, which is more obvious in the translation of long sentences. The scheme proposed in this paper is simple and effective and does not require an additional complex structure, only the addition of a residual link module and an additional loss module. Moreover, the label information is well retained through the residual link module during the training process, and the effect will not decrease significantly, even in the face of long sentences; (c) The class method is a hard decoding method, which will affect the translation of non-decoding constrained dictionary words, thus affecting the fluency of sentences. In addition, the training time of this decoding method is obviously longer. The decoding time of the proposed scheme is close to that of the original model, and the overall translation quality is better than that of the hard decoding method. The experimental results show that the proposed method is effective and superior.

## 3. Transformer-Based Neural Machine Translation Model

Transformer [32] is a neural machine translation model based on the attention mechanism proposed by the Google team in 2017. The model completely solves the problem that the recurrent neural network is restricted by cyclic computing and has achieved remarkable results in machine translation, becoming the mainstream network architecture of neural machine translation. This paper selects the Transformer model to carry out experiments. The content of Section 3.1 is the introduction of the traditional Transformer model. The content of Section 3.2 is the introduction of the improved Transformer model in Scheme 2 and Scheme 3 proposed in this paper.

### 3.1. Traditional Transformer

The Transformer uses an encoder-decoder structure, as shown in Figure 1. The encoder and decoder in the model are both N = 6 layers; each layer has the same structure, but the parameters are not shared.

Each layer of the encoder part contains two sub-layers, followed by the self-attention sub-layer and the feedforward network sub-layer. The attention mechanism in the Transformer is formalized as:

$$Attention(Q, K, V) = softmax\left(\frac{QK^\mathsf{T}}{\sqrt{d_k}}\right)V, \tag{1}$$

where $Q \in \mathbb{R}^{n_q \times d_k}$, $K \in \mathbb{R}^{n_k \times d_k}$, and $V \in \mathbb{R}^{n_k \times d_v}$; the self-attention mechanism is essentially the case of $Q = K = V$ in Formula (1).

Each sub-layer has residual connection and layer normalization operation:

$$LayerNorm(x + Sublayer(x)), \tag{2}$$

where $Sublayer(x)$ represents the self-attention sublayer or the feedforward network sublayer.

The output dimensions of all sub-layers in the network are equal. Each layer of the decoder part contains three sub-layers. A cross-attention sub-layer is added between the self-attention sub-layer and the feedforward network sub-layer. The sub-layer query is the output from the self-attention sub-layer. The keys and values are the output of the encoder

so that each position in the decoder can perform attention calculations on all positions in the encoder. The self-attention sub-layer of the decoder is different from the encoder. It adds a mask mechanism so that each position in the decoder can only calculate the left and the current position to ensure that only the current position and the left side have been generated when predicting the next word. The Transformer uses a parameter-free functional position encoding to preserve the position of words in the sequence to identify the order relationship in the language:

$$P_{(p,2i)} = \sin\left(p/10,000^{2i/d_{model}}\right), \tag{3}$$

$$P_{(p,2i+1)} = \cos\left(p/10,000^{2i/d_{model}}\right), \tag{4}$$

where $p$ is the position and $i$ is the dimension subscript. This functional position encoding can also enable the model to generalize to a sequence length not seen in the training process [32].

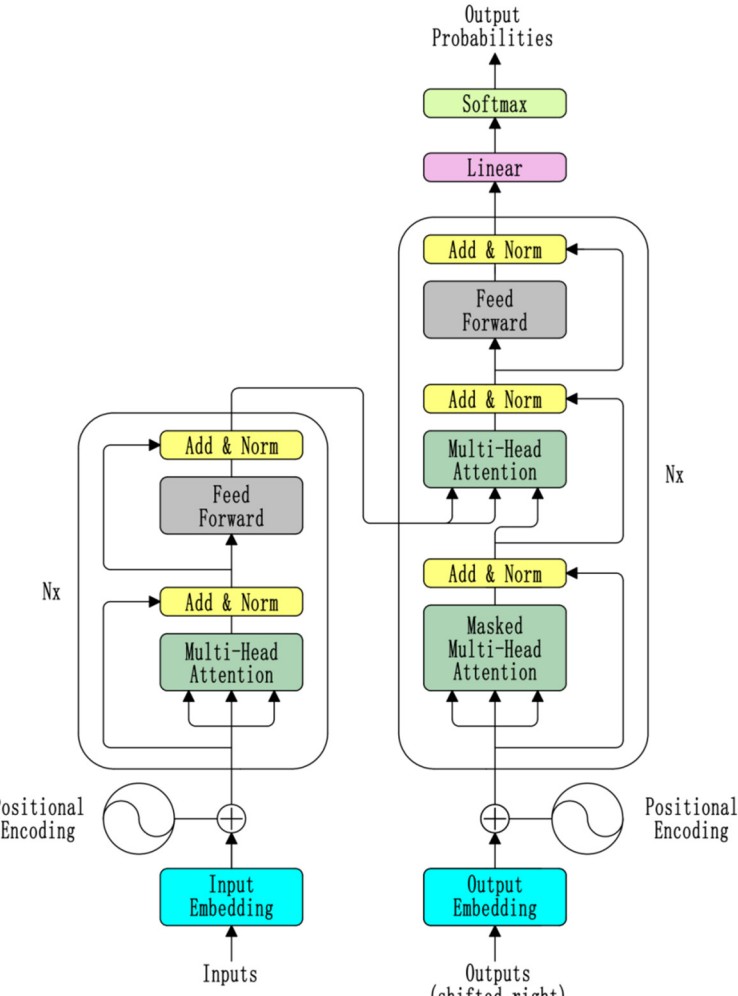

**Figure 1.** Transformer structure.

### 3.2. The Improved Transformer in Our Scheme

For the source language sentence, the vector representation of the sentence input to the encoder end is obtained by adding the word embedding operation and the position encoding. Since the encoder is a multi-layer stacked structure, it is easy to cause the multi-layer units inside it to be like the Elmo [33] model. The information contained in the output vectors of each layer will have different emphasis on syntax and meaning. The units

near the top layer will focus more on grammatical information. The annotation information of the external dictionary and the semantic information of the term itself will be lost. Therefore, this paper adds a residual connection structure to the encoder end, as shown in Figure 2.

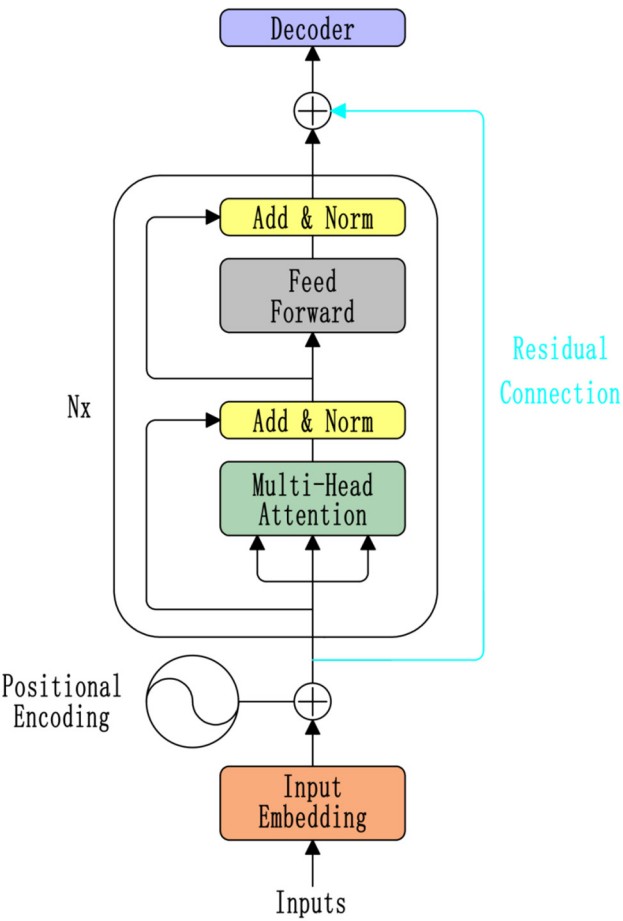

**Figure 2.** Residual connection encoder.

The residual connection structure fuses the source language word vector that adds the position encoding with the output vector of the last layer on the encoder end, as shown in Equation (5).

$$C_{out} = C_{l6} + C_{emb}, \tag{5}$$

where $C_{out}$ is the output vector of the encoder, $C_{l6}$ is the output vector of the sixth layer of the encoder, and $C_{emb}$ is the output vector of the embedding layer.

The residual connection structure is added to ensure that the word vector input decoder contains the term phrase dictionary label information and word meaning information. Experiments show that the improved model improves the translation quality of term words. On the decoder side, in order to make the model adapt to the Character&Term scheme proposed in this paper, a term word additional loss module is added, as shown in Figure 3. This module encourages the translation model to focus on the translation of terminologies and similar methods have been proven to be effective in Reference [24]. Specifically defined as:

$$J(\theta) = arg\max_{\theta}\{P(y|x;\theta) + \lambda \times P(b|x;\theta)\}, \tag{6}$$

where $y$ is the target reference translation, $b$ is the term word sequence generated by using the Character&Term scheme. $y$ is set as a hyper-parameter of 0.5 in the comparative exper-

iment of this paper. No new parameters are added when the loss module is introduced, which only affects the loss calculation during the training of the standard NMT model.

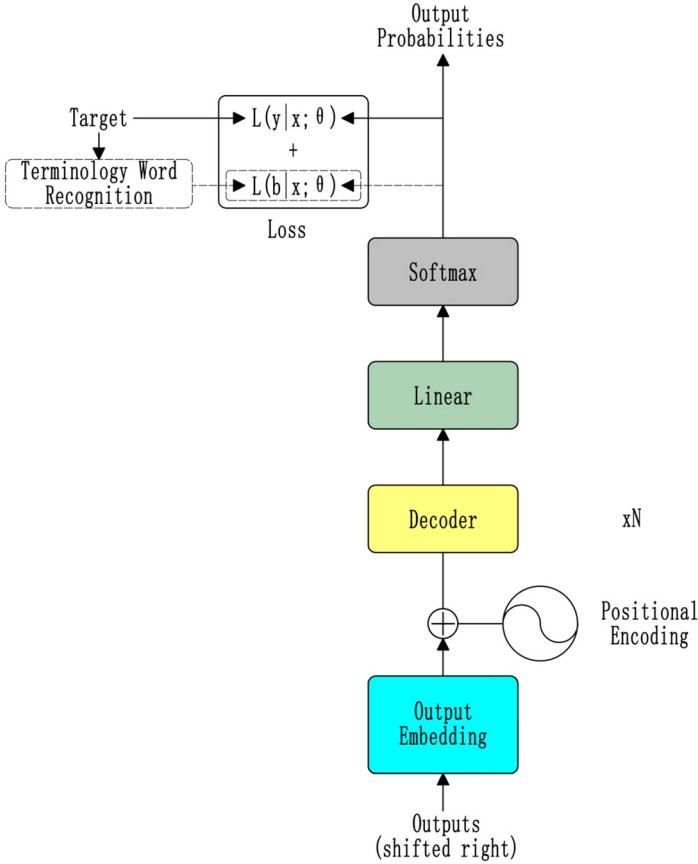

**Figure 3.** Terminology word additional loss module.

## 4. Scheme

In order to solve the problem of terminology translation, this paper constructs a terminology dictionary in the electrical field and uses the dictionary information to impose constraints on the corpus at the source language and the target language. In addition, a residual connection structure is added to the encoder part of the model to make the constraint information better utilized. Based on this, according to different use environments, this paper proposes three schemes to make dictionary knowledge effectively constraint model.

### 4.1. Scheme 1: Replace the Source Language Term Phrase (B&O-Replace)

The scheme replaces the corresponding English terms in the source language with Chinese terms in the term dictionary and labels <B><O> at the beginning and end of the target language phrase in the source language to realize the constraint of dictionary information. For example, in the source language sentence 'Clearances, creepage distances and solid insulation.', there are terms 'creepage distances' and 'solid insulation', which have their corresponding interpretations for '爬电距离' and '固体绝缘' in the terminology dictionary. The source language sentence after term substitution becomes 'Clearances, <B>爬电距离<O> and <B>固体绝缘<O>.'. It is worth noting that the scheme only imposes constraints at the data level and has a good effect. It can be implemented in any field and any scene when using any translation model.

### 4.2. Scheme 2: Use Residual Connection at Encoder after Term Phrase Replacement (Res + B&O-Replace)

On the basis of Scheme 1, this scheme improves the mainstream translation model Transformer to make better use of constraint information. The specific model improve-

ment method is shown in Section 3.2. Experiments in this paper show that this scheme significantly improves the translation accuracy of term words and can be used in any field and any scene.

### 4.3. Scheme 3: The Target Language Uses Dictionary Constraints at the Same Time (Character&Term)

The scheme is designed for English–Chinese machine translation scenarios. Since the current word segmentation method will cause a lack of term information on the Chinese side, dictionary information is used to constrain the Chinese side during word segmentation. Specifically, in the target language sentence '电气间隙、爬电距离和固体绝缘', '爬电距离' and '固体绝缘' are the term words in the dictionary, and the word segmentation result after the constraint is '电\0气\0间\0隙\0, \0爬电距离\0 and \0固体绝缘'. Among them, '\0' represents a space character and will not be used as a label information input model. In order to enhance the term constraint effect of the target language, we add an additional loss module at the end of the decoder. This scheme significantly improves the BLEU value of the translation results in the English–Chinese machine translation scenario.

## 5. Experiment

The experiments in this paper are all based on the Transformer model. By designing multiple sets of different data preprocessing experiments and model comparison experiments, the effectiveness of the scheme described in this paper is verified.

### 5.1. Data Preprocessing

In order to realize machine translation in the field of electrical engineering, the data sets of all experiments in this paper adopt Chinese and English parallel corpus in the field of electrical engineering. The content mainly comes from some Chinese and English materials collected in the field of electrical engineering, including some professional books and documents in the field of electrical engineering, some related technical forums and official websites in the field of electrical engineering. The training set used in the experiment has about 190,000 bilingual parallel corpora; the validation set and the test set each have 2000 bilingual parallel sentence pairs. Table 1 shows an example of the parallel sentence pairs in the corpus.

**Table 1.** Examples of parallel sentence pairs.

| Source | Target |
|---|---|
| The electronic device comprises an electrical storage device. | 所述电子装置包括一个蓄电装置。 |
| Pulsed magnetic flux leakage field testing technology based on 3D magnetic field analysis. | 基于三维场测量的脉冲漏磁检测技术。 |
| Transient voltage stability of independent electric power systems with induction motors. | 含感应电动机的独立电力系统暂态电压分析。 |

The external knowledge used in the experiment is a dictionary containing 38,859 pairs of bilingual term phrases in the electrical field. The term pairs in the dictionary are included in the training set, test set and verification set. The dictionary is established by a dictionary containing 200,000 pairs of bilingual term words in the electrical field. The specific screening steps are as follows: 1. Screen out the English term words in the form of phrases. Because the research in this paper aims at the problem of phrase mismatch and improper translation caused by word-by-word translation of English term phrases in English–Chinese machine translation, only the English phrase part in the dictionary is retained to construct a phrase dictionary; 2. Screening out the term words that appear in the target language sentences in the Chinese side of the dictionary. Table 2 shows the examples of term phrases in the phrase dictionary.

**Table 2.** Example of term phrases.

| Source | Target |
|---|---|
| induction motor | 感应电动机 |
| electrical system | 电力系统 |
| pump storage unit | 抽水蓄能机组 |

When the term phrase dictionary information is integrated into the English side of the source language, the phrases contained in the dictionary of the source language in the parallel sentence pairs are replaced with their corresponding target language phrases, and the tags <B><O> are marked at the beginning and end of the target language phrases on the source language side. Finally, the nltk tool is used to segment the word and input the model. The target language side is processed in two different ways for comparative experiments. In the baseline model and control group, the target language was processed using the Jieba tool. In order to increase the model's recognition of the target language term words, this paper designs a word segmentation method that combines word segmentation and the use of a term dictionary, and the sentence parts outside the target language term words are input into the model by word segmentation. Table 3 shows the parallel sentence pairs processed in different ways.

**Table 3.** Examples of parallel sentence pair processing results.

| Source | EN | Clearances, Creepage Distances and Solid Insulation. |
|---|---|---|
| B&O-Replace | EN | Clearances, <B>爬电距离<O> and <B>固体绝缘<O>. |
| Target | ZH | 电气间隙、爬电距离和固体绝缘 |
| Jieba | ZH | 电气\0间隙\0、\0爬\0电\0距离\0和\0固体\0绝缘 |
| Character&Term | ZH | 电\0气\0间\0隙\0、\0爬电距离\0和\0固体绝缘 |

Source is the source language sentence, B&O-Replace is the source language term phrase replacement method proposed in this paper, Target is the target language sentence, Jieba means using Jieba word segmentation tool for word segmentation, Character&Term is the word segmentation method proposed in this paper, which combines word segmentation and the use of term dictionary. It should be noted that the term Clearances in English sentences are individual words; there is no mismatch problem. Compared with term phrases, it will have a better model learning effect, so it is not included in the scope of annotation. The $'\backslash 0'$ represents a space character that will not be used as a label information input model. The encoder structure designed in this paper is adopted, and the training process after processing the data with B&O-Replace and Character&Term is shown in Figure 4.

### 5.2. Experimental Settings

In this paper, the open-source NMT system OpenNMT is used to implement the baseline model Transformer. In terms of data set processing, the length of sentences in the corpus is limited to 100; that is, sentences with a length of more than 100 will be filtered. The vocabulary size is set to 60,000, and the shared vocabulary is used. Words that are not in the vocabulary are represented by <UNK>. During the training process, both the word vector dimension and the hidden layer dimension inside the codec are set to 512, the batch_size is set to 64, and the number of multi-head attention heads h is set to 8. The Adam optimization algorithm is used, and the neuron random deactivation probability (dropout) is set to 0.1. A total of 30,000 steps are trained in this experiment, and the model is verified every 2500 steps. The beam search method is used in decoding, where beam_size is set to 5, the hyper-parameters $\lambda$ in the additional loss module are set to 0.5 according to experience, and the remaining parameters use the default parameters of OpenNMT.

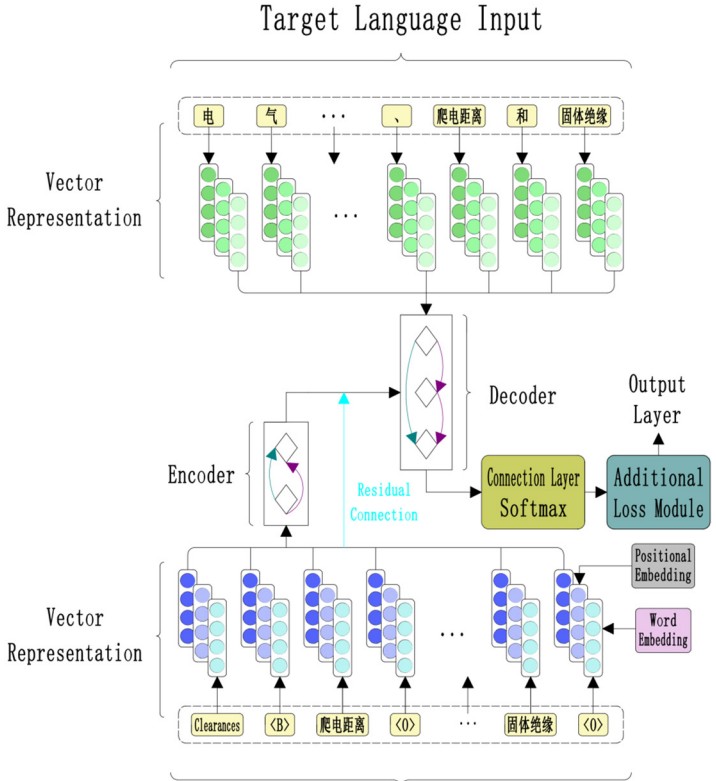

**Figure 4.** Training process.

All parameters in the experiment are consistent, and the BLEU value of the translation results and the correct translation rate of the term words are comprehensively evaluated. The 2000 sentence pairs in the test set contain 1626 term words in the term phrase dictionary. The correct translation rate of term words is calculated as the percentage, and the number of correct term translations generated in the output accounts for the total number of terms in the test set. Specifically, as shown in Formula (7), $P(b)$ is the translation accuracy of terminologies, $n(x_b)$ is the number of terminologies in the translation result and $n(y_b)$ is the number of terminologies in the reference answer.

$$P(b) = \frac{n(x_b)}{n(y_b)}. \tag{7}$$

*5.3. Comparative Experiment*

Baseline model [32]: Using the original Transformer structure, neither the source language nor the target language is marked.

Wu et al. [20]: A method of fusing source language context features. We extract the context features of the term words in the term dictionary according to the experimental steps of Wu et al., use '1 prev, 1next' as the context vector, and use the best C mode proposed by Wu for fusion. The experimental settings are the same as Wu. The number of embedding dimensions, feedforward layer dimensions, and encoder modules is 512, 2048 and 6, and the source language and the target language do not do any other preprocessing.

Xiong et al. [12]: A method of incorporating label information into the model. We label the term phrase dictionary according to the method of Xiong et al. and adopt the third fusion method proposed by Xiong to fuse the label information in the additional embedding layer with the model. Neither the source language nor the target language perform any other preprocessing.

Hu et al. [21]: A method of fusing key information by adding additional encoders. We follow the method of Hu et al. and integrate the terms in the term dictionary as key information into the model by adding an encoder. The source language and the target language do not do any other preprocessing.

### 5.4. Experimental Results

We focused on the problem of phrase mismatch and improper translation caused by word-by-word translation of English terminologies in English–Chinese machine translation in the field of electrical engineering to carry out the experiment. The source language and the target language used different ways to integrate the knowledge of the terminologies phrase dictionary and changed the encoder structure to use the dictionary information better. Three schemes of integrating dictionary knowledge were proposed: B&O-Replace, Res + B&O-Replace, Character&Term. The experimental results are shown in Table 4 ("↑"represents an improvement over the baseline model).

**Table 4.** Experimental Results.

| Model | BLEU/% | Term/% |
|---|---|---|
| Baseline | 32.2 | 46.8 |
| Wu et al. [21] | 33.05 | 57.6 |
| Xiong et al. [13] | 31.98 | 40.3 |
| Hu et al. [22] | 32.56 | 51.7 |
| B&O-Replace | 32.3(+0.1) ↑ | 52.5 |
| Res+ B&O-Replace | 33.04(+0.84) ↑ | 91.3 |
| Character&Term | 35.8(+3.6) ↑ | 95.1 |

The results show that the BLEU values of the three proposed schemes are 0.1, 0.84 and 3.6 higher than the baseline model, and the correct rate of terminology is 5.7%, 44.5% and 48.3% higher than the baseline model. It is proved that the three schemes have improved the BLEU and term accuracy on the baseline model and are competitive in the latest model proposed by peers.

### 5.5. Results Analysis

In order to intuitively show the effectiveness of the proposed scheme, Figures 5 and 6 show the differences in the translation results of different models in the test set with BLEU and the correct translation rate of term words as ordinates.

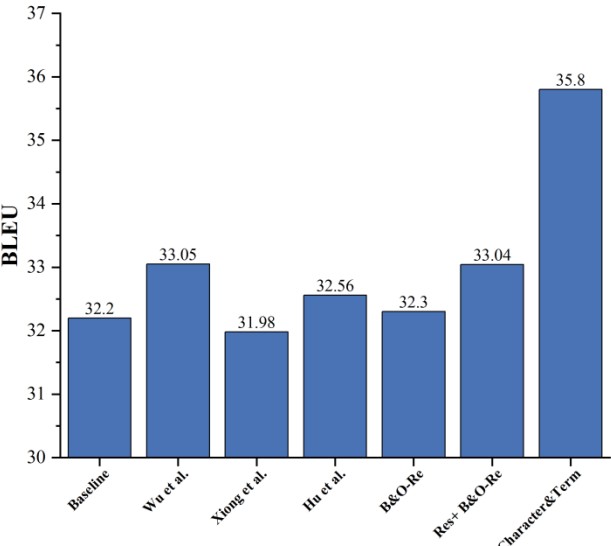

**Figure 5.** BLEU value comparison [13,21,22].

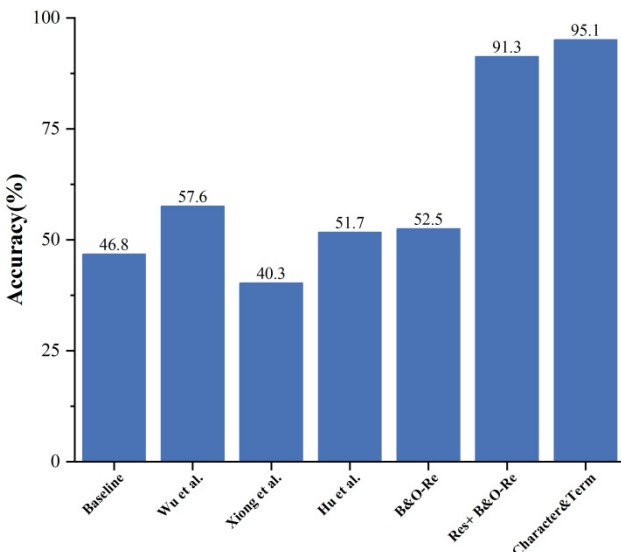

**Figure 6.** Comparison of correct translation rate of terminological words [13,21,22].

It can be seen from Table 4 and Figure 5 that when the B&O-Replace scheme is used for term dictionary knowledge fusion, the BLEU value of the translation result is 0.1 higher than that of the baseline model, indicating that the marking method used in this paper can effectively integrate external dictionary knowledge with the model. When the Res + B&O-Replace scheme is used for term dictionary knowledge fusion, the BLEU value of the translation result can be increased by 0.84 compared with the baseline model, which is greater than the improvement of the B&O-Replace scheme. It shows that the residual connection structure is added to the top layer of the word embedding vector and the encoder with the label, which can effectively integrate the information carried by the two. It not only ensures the integrity of the top-level grammatical information but also ensures the full use of the underlying semantic information so that the semantic information in the label is more fully learned by the model and the translation result is more accurate. When using the Character&Term scheme, the BLEU value of the translation result is increased by 3.6, indicating that under the premise of integrating prior knowledge in the source language, the method of integrating prior knowledge in the target language can further improve the symmetry of the encoder and decoder information, thereby improving the utilization rate of prior knowledge.

From the analysis of Table 4 and Figure 6, it can be seen that when the B&O-Replace scheme is used for term dictionary knowledge fusion, the accuracy of the translation results in the test set is 5.7% higher than that of the baseline model, indicating that the model can effectively learn the word meaning information in the label after adding the tagged external dictionary information to the source language, thereby improving the translation accuracy of the term words. When the Res + B&O-Replace scheme is used for term dictionary knowledge fusion, the term accuracy in the translation results of the test set reaches 91.3%, which is 44.5% higher than that of the baseline model, indicating that adding a residual connection structure between the tagged word embedding vector and the top layer of the encoder can make the word meaning information in the label more fully learned by the model. The scheme improves the term accuracy by 38.8% compared with the B&O-Replace scheme without residual connection structure. It proves that using the residual connection structure to directly input the label information into the decoder can more effectively improve the utilization of label information. At the same time, it shows that after the six-layer encoder, the label information of the embedded layer will be lost to a certain extent. When using the Character&Term scheme, the term accuracy rate is 95.1%, which is a similar result to the scheme Res + B&O-Replace but still 3.8% higher than the former. It

shows that the fusion of the target language term label and the additional loss module at the decoder side can further improve the accuracy of the terms in the translation results.

In the comparison experiment, the methods of Wu et al. and Hu et al. have improved the BLEU value and the accuracy of term words compared with the baseline model, but the effect is lower than the scheme proposed in this paper. The BLEU value and term accuracy of Xiong et al.'s method are lower than those of the baseline model. We speculate that it may be due to the difference between English phrases and Chinese words. When the English end is input into the embedding layer according to the space word segmentation, the phrase will be assigned to more than one vector, which leads to the fact that the source language term phrase cannot be well aligned with the label information during embedding, resulting in a mismatch of information.

### 5.6. Pretreatment Comparison Experiment

Dong et al. [10] and Xiong et al. [12] both used the preprocessing method of direct splicing in the corpus. The preprocessing method used in this experiment is to replace the corresponding position of the source language after the target language term words are labeled. The replacement method not only takes into account the research experience of Dong et al. but also combines the language characteristics of electrical terms. Here, in order to prove the superiority of the preprocessing scheme we adopted, a comparative experiment with the direct stitching method was designed. When the preprocessing method of direct splicing is adopted, the results of corpus processing are as shown in Table 5.

**Table 5.** Preprocessing method of direct splicing.

| Source | EN | Clearances, Creepage Distances and Solid Insulation. |
|---|---|---|
| B&O-Splice | EN | Clearances, creepage distances <B>爬电距离<O> and solid insulation <B>固体绝缘<O>. |
| Target | ZH | 电气间隙、爬电距离和固体绝缘 |

We use the pre-processed corpus to replace the corpus in the three proposed schemes, and the experimental results are shown in Table 6 ("↓" represents a decrease compared to the baseline model).

**Table 6.** Pretreatment comparison experimental results.

| Model | BLEU/% | Term/% |
|---|---|---|
| Baseline | 32.2 | 46.8 |
| B&O-Replace | 32.3 | 52.5 |
| B&O-Splice | 31.02 (↓) | 40.6 (↓) |

The results show that when the prior knowledge of the term dictionary is directly spliced at the corresponding position of the source language, the BLEU value and the term accuracy will be reduced, and the result is lower than the preprocessing scheme used in this paper. The reason for the analysis may be that the source language end of the dictionary used in this paper is a phrase with more than 1 English word. Direct splicing will affect the use of prior knowledge and the sentence structure of the source language sequence.

### 5.7. Hyper-Parameter $\lambda$ Fine-Tuning and Ablation Experiment

In order to adapt to the fusion of terminologies in the target language in Scheme 3, an additional loss module with hyper-parameter $\lambda$ is added in this paper. Figures 7 and 8 are the changes of BLEU and the correct rate of term words when the hyper-parameter T is fine-tuned. When $\lambda$ is 0, it means that no additional loss module is used. Obviously, when $\lambda$ increases from 0 to 0.5, the BLEU value and the correct rate of term words are improved. The BLEU value is increased by 0.53, and the correct rate of term words is increased by 1.5%,

which proves that the additional loss module improves the translation performance of the model. When $\lambda$ is greater than 0.5, the BLEU value gradually decreases, which indicates that over-biased translation of term words will lead to a decrease in the ability of the model to translate non-term words, which is consistent with Chen et al. [24]. At the same time, as $\lambda$ becomes larger, the accuracy of term words is stable at 94.9–95.2%. Therefore, when setting the test, we take 0.5 as the value of $\lambda$.

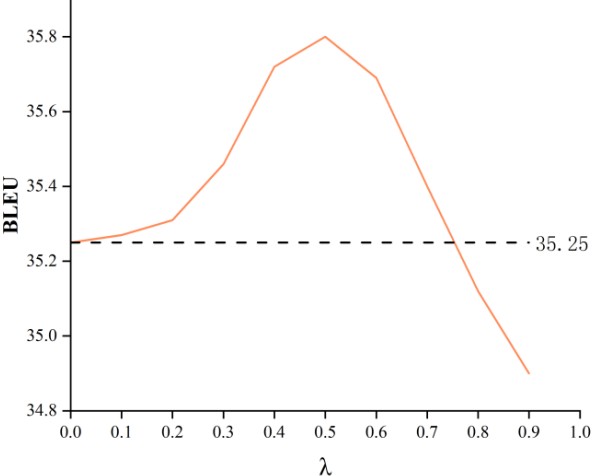

**Figure 7.** The influence of hyper-parameter $\lambda$ on BLEU.

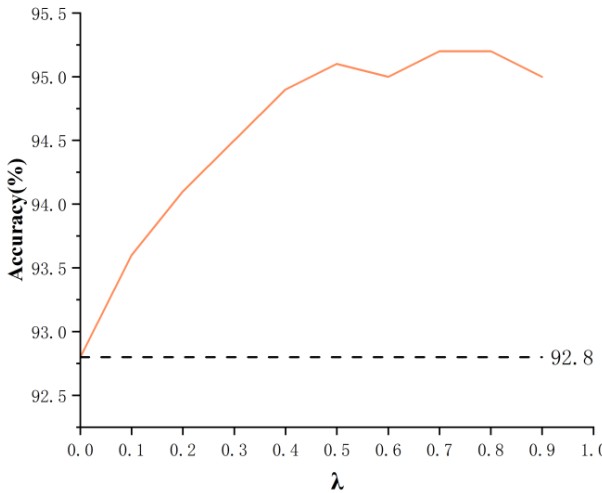

**Figure 8.** The effect of hyper-parameter $\lambda$ on the accuracy of term words.

### 5.8. Example Analysis

Some test set reference sentences, baseline model translation results and several model translation results integrated with term phrase dictionary knowledge are compared, as shown in Table 7. Compared with the baseline model, the translation results of B&O-Replace, Res + B&O-Replace and Character&Term, which integrate the knowledge of terminology phrase dictionary, are not only more fluent but also reduce the <UNK> words, and the term words in the sentence are well translated. All three schemes can correctly translate the unknown term 'rectification unit' in the baseline model, which proves the effective integration of prior knowledge. In the translation results of the Res + B&O-Replace scheme, the terms 'combined power generation system' and 'heater' that cannot be translated by the baseline model are translated into synonyms 'hybrid power generation system' and 'thermal power generating unit'. In the translation results of the Character&Term scheme, the terms 'solar photovoltaic' and 'heater' that cannot be translated by the baseline model

are accurately translated. This shows that the proposed scheme improves the translation performance of the model on the basis of effectively translating the term words in the dictionary, which is reflected in the higher overall quality of the translation results.

**Table 7.** Examples of translation results.

| Source | Modeling & Simulation of Generator Excitation System Rectifier Unit Based on Simulink | The Configuration Research of the Battery and Fuel Cell in Hybrid Power Generation System of Photovoltaic and Fuel Cell | The Quantitative Analysis on the Steam Extraction Pressure Loss Deviated from the Target on the Thermal Efficiency |
|---|---|---|---|
| Target | 基于Simulink的发电机励磁系统整流单元的建模与仿真 | 太阳能光伏-燃料电池联合发电系统蓄电池和燃料电池的配置研究 | 加热器抽汽压损偏离目标值对机组热经济性影响的定量分析 |
| Baseline | 基于Simulink的发电机励磁系统<unk>的建模与仿真 | 风光<unk>中蓄电池与燃料电池的结构研究 | 电厂抽汽<unk>对热效率影响的定量分析 |
| B&O-Replace | 基于Simulink的发电机励磁系统整流单元建模与仿真 | 太阳能和燃料电池混合<unk>中电池和燃料电池的配置研究 | 热效率目标对蒸汽抽汽<unk>的定量分析 |
| Res + B&O-Replace | 基于Simulink的发电机励磁系统整流单元建模与仿真 | 光伏-燃料电池混合发电系统中蓄电池和燃料电池的配置研究 | 火力发电机组抽汽压损对机组热效率影响的定量分析 |
| Character&Term | 基于Simulink的发电机励磁系统整流单元建模与仿真 | 太阳能光伏和燃料混合发电系统中蓄电池与燃料电池的配置研究 | 加热器抽汽压损偏离时对机组热效率影响的定量分析 |

*5.9. Comprehensive Analysis*

(1) Integrating prior knowledge into low-resource and professional domain corpus is an effective way to improve translation quality. Experiments show that the three schemes we proposed can improve the BLEU value and term accuracy of machine translation;

(2) Comparing the three schemes, the BLEU value is the most improved in the third scheme. The reason is that the source language end is integrated with the prior knowledge, and the target language end is also integrated with the same prior knowledge, which effectively prevents the ambiguity and forgetting of the label knowledge after the six-layer encoder of the model. In terms of term accuracy, both Scheme 2 and Scheme 3 have a greater improvement than the baseline model, which indicates that when prior knowledge is added in the preprocessing stage, after the multi-layer attention of the model, the prior knowledge will be lost, and our improvement in the model effectively prevents this loss; in terms of the translation results of the test set, Scheme 1, Scheme 2 and Scheme 3 can effectively translate the term words that cannot be translated by the baseline model, and the effect of Scheme 2 and Scheme 3 is better than that of Scheme 1, which further shows that the improvement of the model improves the utilization of prior knowledge;

(3) The results of comparative experiments show that different types of terms require different preprocessing methods to adapt to the translation model. For example, the terms of a single word, term phrases, and professional expression sentences of terms need to be classified or customized to obtain better translation results. This will be our future research direction of interest.

## 6. Conclusions

This paper constructs a term phrase dictionary in the field of electrical engineering and proposes three schemes to use dictionary knowledge to constrain translation models. In the first scheme, <B><O> markers are used to replace terms in the source language. Compared with the baseline model, the BLEU value of the translation result is increased by 0.1, and the accuracy of term words is increased by 5.7%. The advantage of this scheme is that it is simple and easy to implement and can be implemented in any model. In the second scheme, <B><O> markers are used to replace terms at the source language end, and a residual connection structure is added at the encoder end. Compared with the baseline

model, the BLEU value of the translation result is increased by 0.84, and the accuracy of term words is increased by 44.5%. The scheme achieves a correct translation rate of 91.3% of term words. On this basis, the model can also obtain a higher overall quality translation. The third scheme uses the segmentation method of combining characters and terms on the Chinese side and adds an additional loss module. Compared with the baseline model, the BLEU value of the translation result is increased by 3.6, and the correct rate of term words is increased by 48.3%. This method can better improve the quality of the translation when the target language is Chinese. The experimental results and translation examples strongly prove the effectiveness of the proposed scheme. By integrating the term phrase dictionary, the problem of phrase mismatch and improper translation caused by word-by-word translation of English term phrases is alleviated.

We will build a better terminology dictionary in the electrical field in future work, which contains various forms of terms, constantly improve the methods of using dictionary knowledge, and explore effective ways to further improve neural machine translation in the professional field.

**Author Contributions:** Research conceptualization: Z.W.; Model building: Z.W.; Data collection: Z.W. and Y.C.; Experiment design: Z.W., Y.C. and J.Z.; Manuscript preparation: Z.W.; Manuscript review: Z.W., Y.C. and J.Z. All authors have read and agreed to the published version of the manuscript.

**Funding:** This research was funded by Juwei Zhang, grant number U2004163, and the APC was funded by Juwei Zhang.

**Institutional Review Board Statement:** Not applicable.

**Informed Consent Statement:** Not applicable.

**Data Availability Statement:** The data sets used and/or analyzed during the current study are available from the corresponding author upon reasonable request.

**Conflicts of Interest:** The authors declare no conflict of interest.

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
