# Peer review of "Machine Translation of Electrical Terminology Constraints"

_information, doi:10.3390/info14090517_

Round 1
Reviewer 1 Report
The aim of the paper is to propose methods to integrate the dictionary knowledge into the machine translation model.
The paper is written in the scientific style, containing new and significant information adequate to justify publication. The relevant literature is cited showing understanding of the research domain. The methodology, the research and results are presented and analyzed. New research results are explained and implications given for practice and further research.
The paper is suggested for publication, how with suggestions for some minor corrections to improve readability:
Light language editing is needed (e.g. correct blanks before comma, verify „In the word“ or „In the world“?
Explain before the first mention: <B><O>, <UNK>
Suggestion to add few more citations related to NMT research: Gašpar et al. (2022) Measuring Terminology Consistency in Translated Corpora: Implementation of the Herfindahl-Hirshman Index – for possible further research of terminology consistency in translated corpora; Evaluation of NMT for low-resourced languages: DunÄ‘er et al. (2020) Automatic Machine Translation of Poetry and a Low-Resource Language Pair; Seljan et al. (2020) Human Quality Evaluation of Machine-Translated Poetry
Light language editing
Author Response
Thank you very much for taking the time to review this manuscript. Please find the detailed responses below and the corresponding revisions. The specific modifications to the article have been highlighted in red font.
Comments 1: Light language editing is needed (e.g. correct blanks before comma, verify „In the word“ or „In the world“?
Response 1: Thank you for your suggestion. We have completely checked the spaces between the punctuation points in the article and corrected the word errors in the article.
Comments 2: Explain before the first mention: <B><O>, <UNK>
Response 2: Thank you for your suggestion. We have provided explanations for<B><O>and<UNK>in the article. The explanation of<B><O>is located in section 4.1 of the article, and specific examples are provided< UNK>provides an explanation in section 5.2 of the article.
Comments 3: Suggestion to add few more citations related to NMT research: Gašpar et al. (2022) Measuring Terminology Consistency in Translated Corpora: Implementation of the Herfindahl-Hirshman Index – for possible further research of terminology consistency in translated corpora; Evaluation of NMT for low-resourced languages: DunÄ‘er et al. (2020) Automatic Machine Translation of Poetry and a Low-Resource Language Pair; Seljan et al. (2020) Human Quality Evaluation of Machine-Translated Poetry
Comments 3: Thank you for your suggestion. We have carefully read and studied the article you provided and cited it in our article.

Reviewer 2 Report
I am missing a better described examined corpus, and/or corpus statistics, e.g. how many terms were found in the source text and in the machine translation etc.
- formal shortcomings: e.g. Hu [22] et al., spaces before punctuation marks etc. If the formula is part of a sentence, the formula should be followed by a full stop or a comma if the sentence continues.
Author Response
Thank you very much for taking the time to review this manuscript. Please find the detailed responses below and the corresponding revisions. The specific modifications to the article have been highlighted in red font.
Comments 1: I am missing a better described examined corpus, and/or corpus statistics, e.g. how many terms were found in the source text and in the machine translation etc.
Response 1: Thank you for raising such a question. We have improved the statistics on the number of terms in the source language corpus and experimental results in the article. Among them, the corpus contains all terminology and phrases, which are explained in Section 5.1 of the article; Our test set has a total of 1626 terminology terms, as explained in section 5.2. In order to make the comparison of experimental results more obvious, we use a percentage (the percentage of correctly translated terms in the 1626 terms in the test set) in the article to indicate the accuracy of the terms.
Comments 2:- formal shortcomings: e.g. Hu [22] et al., spaces before punctuation marks etc. If the formula is part of a sentence, the formula should be followed by a full stop or a comma if the sentence continues.
Response 2: Thank you for raising such a question.We have thoroughly checked and corrected any formatting defects in the article, such as spaces before punctuation. All formulas have been modified according to your suggestion.

Reviewer 3 Report
This paper does not follow academic style of writing and it is very confusing to the readers. For example, in the First line of Introduction section, it says "Machine translation[1]". The reference citation [1] appears without any space. Then, in the fifth line of the introduction there is unnecessary space before the comma "breakthroughs , ". Then again in the second paragraph of the introduction section reference style completely changes to superscript (i.e., for [7], [8]).
The research gap is not clearly presented in the introduction section. Moreover, the contribution of this research is not clearly highlighted.
In the current version of the paper only 2/3 recent papers (i.e., year 2022, 2023) have been cited. Please describe more recent works (i.e., from year 2022, 2023) to better establish the current research gap.
In section 2, clearly highlight the research gap from existing literature in a tabular structure. The current "Related Work" section is confusing to the readers. In this section 2, there are many formatting issues. For example, Bibliography and reference style is inconsistent. Sometimes, the author used References[11][12]. Sometimes, they used "Liu et al." followed by a superscript reference [12]. These inconsistencies throughout the paper is irritating and confusing to the readers.
Figure 1, Figure 2, Figure 3, and Figure 4 uses various colors. Does this color mean anything for differentiating the building blocks? If yes, then please provide legends for each these figures. Otherwise, this is confusing and shows carelessness of the authors.
Almost all the references look wrongly formatted. For example, in reference [26], A. Vaswani (i.e., First Name and Surname) and all on a sudden, in reference [26] it is Peters M (i.e., Surname and First Name). This is completely contradictory formatting. Also, what are [J], [M], [C] etc. abruptly used within the reference sections? These are extremely confusing to the readers and irritating.
Too many formatting issues (space missing when it is necessary, space provided when it is not required, inconsistent reference / citation style)
Author Response
Thank you very much for taking the time to review this manuscript. Please find the detailed responses below and the corresponding revisions. The specific modifications to the article have been highlighted in red font.
Comments 1: For example, in the First line of Introduction section, it says "Machine translation[1]". The reference citation [1] appears without any space. Then, in the fifth line of the introduction there is unnecessary space before the comma "breakthroughs , ". Then again in the second paragraph of the introduction section reference style completely changes to superscript (i.e., for [7], [8]).
Response 1: Thank you for your suggestion. We have made the modifications according to your suggestion. We have checked and corrected all spaces and punctuation in the article, and unified all reference annotations, such as' [1] '.
Comments 2: The research gap is not clearly presented in the introduction section. Moreover, the contribution of this research is not clearly highlighted.
Response 2: Thank you for your suggestion. We have made the modifications according to your suggestion. We highlighted the research contributions of the article in the introduction section and summarized previous work in the relevant work section, explaining the main research gaps.
Comments 3: In the current version of the paper only 2/3 recent papers (i.e., year 2022, 2023) have been cited. Please describe more recent works (i.e., from year 2022, 2023) to better establish the current research gap.
Response 3: Thank you for your feedback. We have recognized the shortcomings of research on the latest related work. We have studied the latest work and compared it with them. They are specifically introduced in the Related Work section of the article. Among them, the references [26,27,28,29,30,32] are newly added projects.
Comments 4: In section 2, clearly highlight the research gap from existing literature in a tabular structure. The current "Related Work" section is confusing to the readers. In this section 2, there are many formatting issues. For example, Bibliography and reference style is inconsistent. Sometimes, the author used References[11][12]. Sometimes, they used "Liu et al." followed by a superscript reference [12]. These inconsistencies throughout the paper is irritating and confusing to the readers.
Response 4: Thank you for your feedback. We recognize the shortcomings in the writing of the second section of the article and have made the following improvements: we have unified the format of reference citations, summarized and classified all work in the form of text at the end of the second chapter, and compared the research gap (due to the length of these contents, we attempted to classify them using a table, but it has poor aesthetics).
Comments 5: Figure 1, Figure 2, Figure 3, and Figure 4 uses various colors. Does this color mean anything for differentiating the building blocks? If yes, then please provide legends for each these figures. Otherwise, this is confusing and shows carelessness of the authors.
Response 5: Thank you for your suggestion. The colors in Figures 1, 2, 3, and 4 in the article do not refer to meaning. Their design is to make the entire table structure clearer and increase the readability of the article.
Comments 6:Almost all the references look wrongly formatted. For example, in reference [26], A. Vaswani (i.e., First Name and Surname) and all on a sudden, in reference [26] it is Peters M (i.e., Surname and First Name). This is completely contradictory formatting. Also, what are [J], [M], [C] etc. abruptly used within the reference sections? These are extremely confusing to the readers and irritating.
Response 6: Thank you for your valuable suggestion. We have made modifications to the format of the references, such as the formatting of surnames and first names, the use of punctuation and spaces, and the removal of markings such as [J], [M], and [C].
Response to Comments on the Quality of English Language
Point 1:Too many formatting issues (space missing when it is necessary, space provided when it is not required, inconsistent reference / citation style).
Response 1: All formatting issues have been revised according to the reviewer's comments.

Round 2
Reviewer 3 Report
The authors have addressed my previous concerns and comments. I have no further comments to add. Good luck to the authors.